# Characteristics of Co-Seismic Surface Rupture of the 2021 Maduo Mw 7.4 Earthquake and Its Tectonic Implications for Northern Qinghai–Tibet Plateau

**Hong Xie [1,2], Zhimin Li [3,*], Daoyang Yuan [4], Xianyan Wang [2], Qi Su [5], Xin Li [3], Aiguo Wang [1] and Peng Su [6]**

[1] Gansu Lanzhou Geophysics National Observation and Research Station, Lanzhou 730000, China
[2] School of Geography and Ocean Science, Nanjing University, Nanjing 210000, China
[3] Qinghai Earthquake Agency, Xining 810000, China
[4] School of Earth Science, Lanzhou University, Lanzhou 730000, China
[5] Department of Geographic Science, Faculty of Arts and Sciences, Beijing Normal University at Zhuhai, Zhuhai 519000, China
[6] Institute of Geology, China Earthquake Administration, Beijing 100036, China
[*] Correspondence: minhero_168@126.com; Tel: +86-971-6124657

**Abstract:** A magnitude ($M_w$) 7.4 Maduo earthquake occurred on 22 May 2021 in the northern Qinghai-Tibet Plateau, with predominantly left-lateral strike-slip faulting and a component of normal faulting within the Bayan Har Block. The co-seismic surface rupture extended in a NWW direction for ~160 km with a complicated geometry along a poorly known young fault: the Jiangcuo Fault. The main surface rupture propagated bilaterally from the epicenter and terminated eastward in horsetail splays. The main rupture can be divided into five segments with two rupture gaps. Field surveys and detailed mapping revealed that the co-seismic surface ruptures were characterized by a series of left-lateral offsets, en echelon tensional cracks and fissures, compressional mole tracks, and widespread sand liquefication. The observed co-seismic left-lateral displacements ranged from 0.2 m to ~2.6 m, while the vertical displacements ranged from 0.1 m to ~1.5 m, much lower than the InSAR inverse slip maximum of 2–6 m. Based on the comprehensive analysis of the causative fault geometry and the tectonic structure of the northern Bayan Har Block, this study suggests that the multiple NWW trending sub-faults, including the Jiangcuo Fault, developed from the East Kunlun fault northeast of the Bayan Har Block could be regarded as the sub-faults of the East Kunlun Fault system, constituting a broad and dispersive northern boundary of the Block, controlling the inner strain distribution and deformation.

**Keywords:** 2021 Maduo Mw 7.4 earthquake; co-seismic surface rupture; Jiangcuo Fault; Bayan Har Block; East Kunlun Fault

## 1. Introduction

At 02:24 am (GMT + 8) local time on 22 May 2021, the moment magnitude and surface wave of a 7.4 earthquake—according to the Global Centroid Moment Tensor (GCM) and the China Earthquake Administration (CEA)—struck the districts around Maduo county in the Northern Tibetan Plateau. The epicenter was located 34.59°N, 98.34°E at a depth of ~10 km (China Earthquake Net Center). It was suggested that the earthquake occurred as the result of E–W trending left-lateral strike-slip faulting with a component of normal faulting, and the focal mechanism plane showed a possible causative fault strike and dip of 92° and 67°, respectively, as well as a −40° rake (https://earthquake.usgs.gov/earthquakes/eventpage/us7000e54r/executive, accessed on 28 June 2022). This earthquake was assigned the highest seismic intensity scale of X from the investigation of the Qinghai Earthquake Agency, making it one of the most intense and strongest earthquakes in China

in recent 20 years after the 2001 Kunlun Mw 7.8 earthquake [1] and the 2008 Wenchuan Mw 7.9 earthquake [2]. Fortunately, the 2021 Maduo earthquake occurred in a broad pasturing and sparsely populated area, where no deaths occurred but seventeen persons suffered minor injuries. Additionally, highway bridges, roads, and walls collapsed in the earthquake.

After the earthquake, geophysical studies were conducted using seismic or geodetic data and InSAR data to simulate the distribution of the surface rupture and faulting kinematics [3–7]. The InSAR-based inversion result revealed that the co-seismic slip developed mainly along the trace of an NWW-trending fault with a maximum slip of 2–6 m [4,6,8–10]. By combining the aftershock distribution and the geophysical results, it was confirmed that the causative fault should be the NWW-trending Jiangcuo Fault (JCF), which spreads east-to-west for approximately ~170 km and has a banding distribution with the strike of 285° [11]. The epicenter and source modeling results also indicated that the rupture was initiated near the Yellow River valley, further propagating about 76 km to the west and 85 km to the east [11].

Although the co-seismic surface rupture distribution, maximum slip, and faulting characteristics after the earthquake have been extensively explored, first-hand field geological and high-resolution field mapping data are still desperately needed to identify the displacements and truly understand the nature of co-seismic strike-slip rupturing structures and kinematic mechanisms. Moreover, there were scarce data related to the causative structures and displacements, as the earthquake occurred on a very poorly known fault in a remote, high mountain region of the Northern Qinghai-Tibet Plateau (average elevation > 4300 m). Coincidentally, our team was conducting geological surveys at Maqin County, which was 100 km from the epicenter. After the earthquake, we arrived at the earthquake region on the same day and began conducting a geological field investigation. Based on the detailed observations and mapping of the field investigation, this study analyzes the co-seismic surface rupture and the displacements associated with the Maduo earthquake and the preexisting geological structure. Furthermore, the regional tectonic significance and the potential future strong earthquake hazards are discussed.

## 2. Tectonic Setting

The continuous collision between the Indian and Eurasian plates results in the expansion and extrusion of the Qinghai–Tibet Plateau to the marginal zone. Within the plateau, five tectonic fault blocks, namely, the Qilian-Qaidam Block, Bayan Har Block, Qiangtang Block, and Lhasa Block, from north to south were shaped by principal strike-slip faults [12] (Figure 1b). The 2021 Maduo earthquake occurred on the Kunlunshankou-Jiangcuo Fault (KLSK-JCF), a secondary fault located in the eastern central area of the Bayan Har Block (Figure 1a). The Bayan Har Block is an elongated wedge-shaped secondary block that forms part of the northern Tibetan Plateau (Figure 1b). It is bounded by the NWW-trending major left-lateral strike-slip Kunlun and SN-trending Minjiang faults in the north, to the southwest by the Ganzi-Yushu-Xianshuihe fault system separating the Qiang Tang Block to the south, and to the southeast by the NE-trending Longmenshan thrust, which also bounds the eastern margin of the Tibetan Plateau (Figure 1a). Since the end of the twentieth century, eight large earthquakes (M ≥ 7) have occurred surrounding the Bayan Har Block: the 1997 Mw 7.5 Manyi earthquake (on the Ganzi-Yushu-Xianshuihe Fault) [13], the 2001 Mw 7.8 Kunlun earthquake (on the Kunlun Fault) [14], the 2008 Mw 7.9 Wenchuan earthquake (on the Longmenshan Fault) [15,16], the 2010 Mw 6.9 Yushu earthquake (on the Yushu-Xianshuihe Fault) [17], the 2013 Mw 6.6 Lushan earthquake (on the Longmenshan Fault) [18], and the 2017 Mw 6.9 Jiuzhaigou earthquake (on the Mingjiang Fault) [19]. The focal mechanism of these earthquakes implies that the Bayan Har Block is currently moving to the southeast relative to the South China block (Figure 1a). Geomorphic observations also suggested that the boundary faults of the Bayan Har Block have been very active since the Holocene. The northern boundary, the East Kunlun Fault, shows an obviously eastward decrease in the left-lateral displacement rates, from >

10 mm/yr across the central Kunlun to <2 mm/yr near the eastern fault terminus [20–25]. To the south rim, the Holocene sinistral strike-slip rate along the Ganzi-Yushu and Xianshuihe faults has been estimated to be 12 ± 2 mm/yr and 11.5 ± 2 mm/yr [26]. As for the southeastern Longmenshan fold-and-thrust belt (Figure 1a), GPS observations confirmed the thrust and dextral shears but with a long-term low deformation rate (<2 mm/yr) [15,27,28]. On the east border of the Block, the Minjiang Fault system consists of the Minjiang and Huya secondary faults. The Minjiang Fault is characterized by thrusting with a shortening rate of less than 2–3 mm/yr, inferred from the rates of the differential rock lift, while the Huya Fault is dominated by left-lateral strike movement with a horizontal slip rate of about 2 mm/yr [27,29].

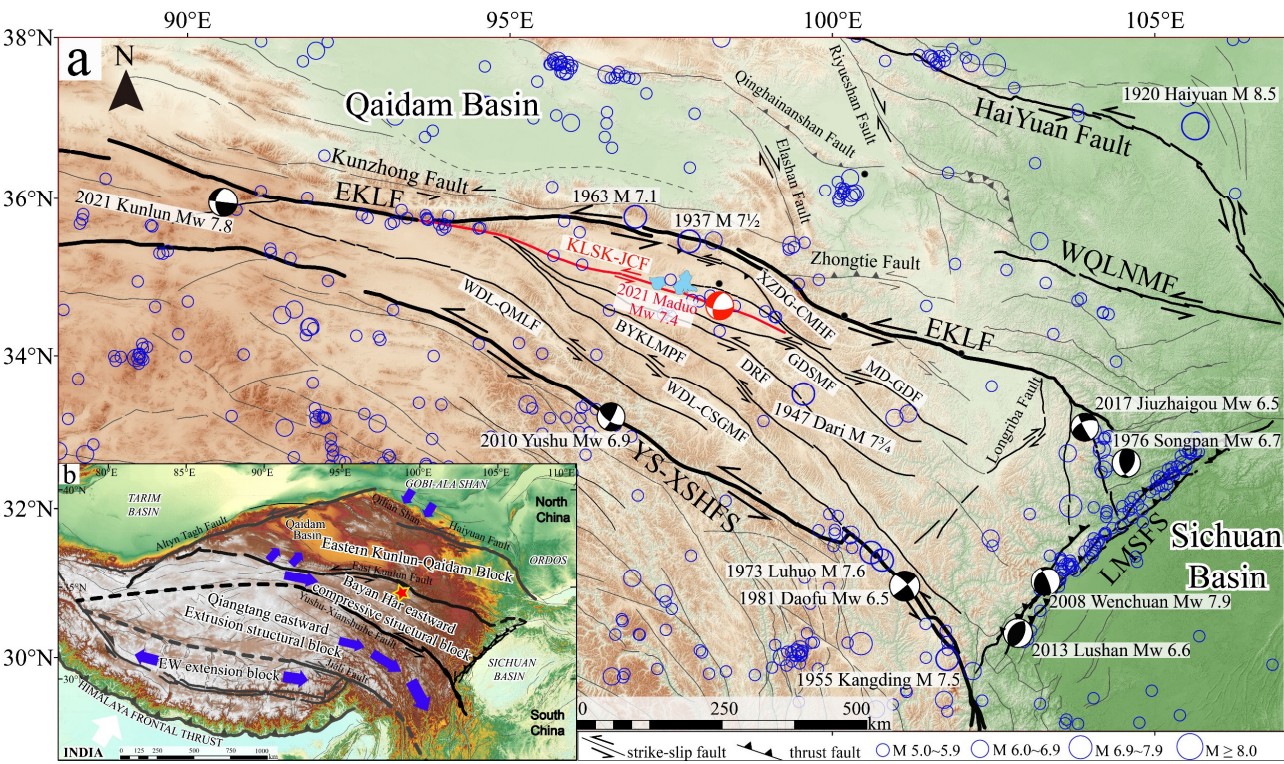

**Figure 1.** (**a**) Seismotectonic map of Bayan Har Block in the northern Qinghai–Tibet Plateau. The active faults were modified from Deng [30]. Epicenters before 1976 were taken from China earthquake network center. Epicenters and Focal mechanism solutions of strong earthquakes (Mw ≥ 6.5) around the Bayan Har Block since 1976 were taken from the Global Centroid-Moment Tensor seismic catalog (https://www.globalcmt.org/). (**b**) Index map of the study area in the Tibetan Plateau. Block boundary was referred from Tapponnier [31]. (EKLF—East Kunlun Fault, WQLNMF—West Qinlin north margin Fault, YS-XSHFS—Yushu-Xianshuihe Fault system, LMTFS—Longmenshan Thrust Fault system, XZDG-CMHF—Xizangdagou-Changmahe Fault, MD-GDF—Maduo-Gande Fault, GDSMF—Gande South margin Fault, BYKLMPF—Bayan Har main peak Fault, WDL-QMLF—Wudaoliang-Qumalai Fault, and WDL-CSGMF—Wudaoliang-Changshagongma Fault).

Besides the major boundary faults around the Bayan Har Block, a sequence of parallel NWW-trending left strike-slip active faults were also developed in the central-to-eastern part of the Block, which are the Maduo-Gande Fault, Jiangcuo Fault, Gande south margin Fault, Dari Fault, Bayan Har main peak Fault, and the Wudaoliang-Changshagongma Fault from north to south (Figure 1a). Little research has focused on these faults because of the remote and hard access. Based on the interpretation of satellite images and a field investigation, Xiong (2010) identified the 50 km long surface rupture of the Maduo-Gande Fault and suggested that the maximum sinistral horizontal and vertical displacements were 7.6 m and 4 m, respectively [32]. Liang (2020) analyzed the surface rupture of the 1947 7¾ earthquake on the Dari Fault (Figure 1b) and suggested the length of 70 km for the surface rupture and a 2–4 m co-seismic horizontal displacement [33]. However, few

studies have been carried out on the seismogenic fault of the Maduo earthquake, the Kunlunshankou-Jiangcuo Fault, and it is a very poorly known fault with no documented large earthquakes.

## 3. Data and Methods

Application of structure-from-motion (SfM) algorithm in photo-based 3D reconstruction provided very high-resolution (cm-scale) but low-cost topography data for characterizing the local geomorphology and measuring fault slip. This technique combines sufficiently overlapping optical images of a target area from various view angles to create point-cloud surface models, and is in turn used to derive high-resolution DEMs and orthophotos [34]. In sparsely vegetated regions such as the study area, the photo-based DEMs can have higher resolution than the LiDAR-derived DEMs. Soon after the earthquake, high-resolution photography and topographical mapping of the surface rupture were conducted by flying across most of the surface ruptures using an unmanned aerial vehicle (UAV) within three days. More than 3000 aerial photos were taken at a mean height of 100 m and yielded an average of 3–5 cm per pixel. In each swath area, the optical images were used to create a digital surface model (DSM) using the SfM technique and a set of ground control points collected with an RTK (real-time kinematics) GPS. Combined with our direct field observations on the ground, the detailed morphology of the surface rupture was obtained prior to its later rapid degradation and erosion.

The rupture trace was investigated and photographed on foot within four days after the earthquake. Displacement vectors were obtained by surveying the ruts in grassland, roads, streams, and other linear features. In addition to our on-ground field observation, orthophotographs and the DSM created from images collected by UAV along the rupture were also analyzed. The typical distinct aerially surveyed swaths were selected, encompassing areas a–e from west to east in Figure 2.

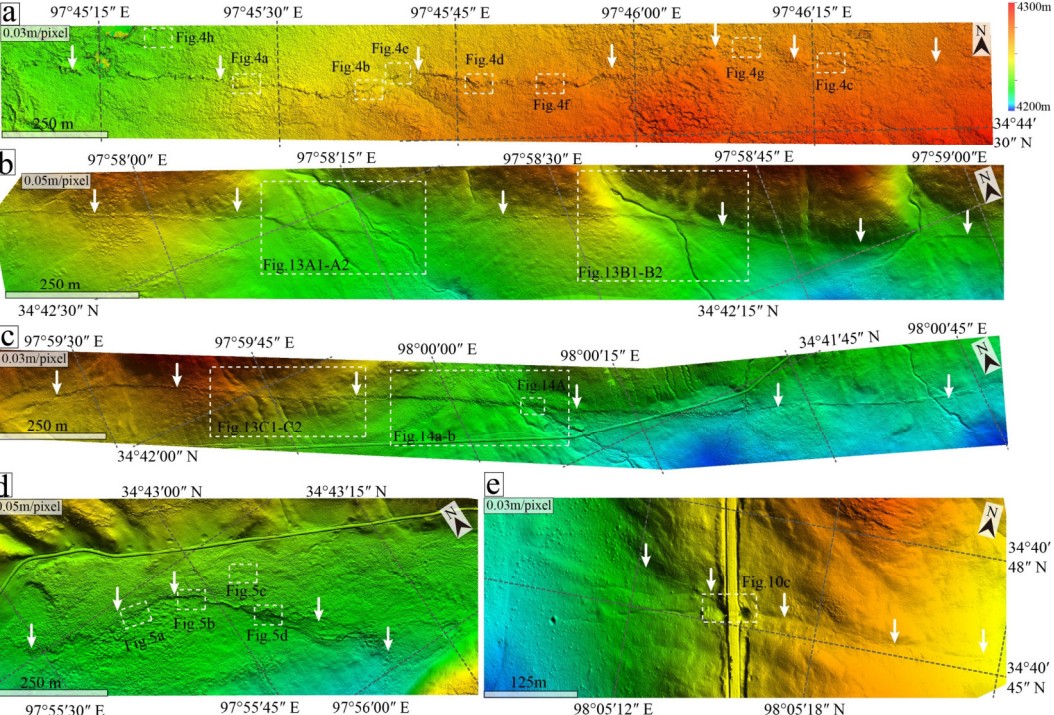

**Figure 2.** High-resolution DSM (3–5 cm/pixel) of typical co-seismic surface rupture of the Maduo earthquake obtained by UAV within three days after the earthquake. (**a–e**) are the aerially surveyed swaths of the DSM from west to east of the co-seismic surface rupture and the locations of Figures 4–14 are also indicated in (**a–e**). The white arrows indicate the co-seismic surface-rupture distribution.

## 4. Results

*4.1. Segmentation of Surface Rupture of the Maduo Earthquake*

According to the comprehensive analysis of the field survey and the comparison of the high-resolution images before and after the earthquake as well as the seismic relocation distribution (Figure 3a,b), the entire surface rupture of the Mw 7.4 Maduo earthquake was determined: it spans a continuous zone of ~160 km long, generally trending NWW, from the south of Eling Lake (97.60°E) to the east across the Yematan Bridge (98.05°E), eastwards to Huanghe town (98.33°E) and further extending to the north of Dongcaoalong lake (98.76°E), further to the dune area of the northern bank of the Yellow River (99.00°E), and ultimately ending in the east of Changma village (99.28°E) (Figure 3d). The co-seismic surface rupture is mainly characterized by a series of shear faults, en echelon tensional or transtensional cracks, shear cracks, and fissures as well as mole track structures, sand liquefaction, and water blasting widely distributed in small valleys and swamp areas. The geometry and the deformation of the Maduo earthquake surface rupture were complicated (Figure 3d,e). According to the different faulting features and the style of the co-seismic surface rupture, it was divided into five segments from west to east, which are the South Eling Lake segment, Yematan segment, Huanghe town segment, Dongcaoalong Lake segment, and Changmahe segment (Figure 3d). The South Eling Lake and Yematan segments are mainly single-stranded and demonstrate continuous left-lateral strike-slip faulting with the remarkable component of a normal dip-slip as well as an intense partial compressional feature (Figures 2, 4, 5 and 6). The middle Huanghe village segment is characterized by a discontinuous surface rupture with pure sinistral strike-slip faulting (Figure 7). The westernmost tip of the surface rupture, the Changmahe segment, diverged into two subsegments as a horsetail exhibiting the typical terminal effect of strike-slip faults. There are two surface rupture gaps between the Huanghe town segment and the Changmahe segment (Figure 3d,e).

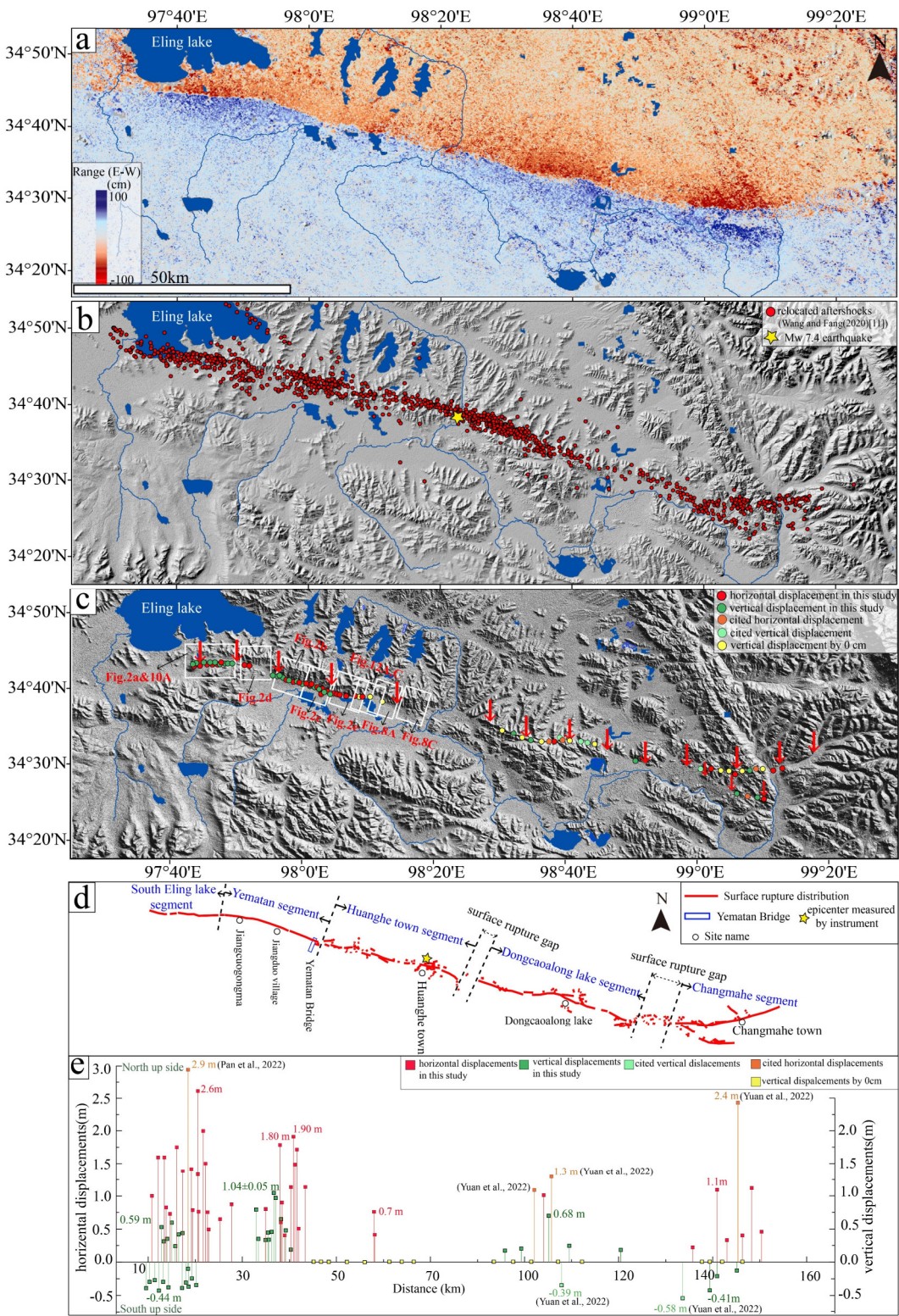

**Figure 3.** (**a**) LiCSAR observations of the Maduo earthquake (Adapted with permission from Dr. Sotirios Valkanioti); (**b**) Relocated aftershocks adapted from Wang et al. (2022) [11]; (**c**) co-seismic surface rupture distribution and the displacement point sites; (**d**) co-seismic surface rupture segmentation; (**e**) displacements' distribution, the cited displacements from Yuan et al. (2022) [35] and Pan et al. (2022) [36].

### 4.2. Characteristics and Morphology of the Co-Seismic Surface Ruptures

South Eling Lake segment: The trace of this segment of the co-seismic surface rupture is quite sharp on the DSM (Figure 2a), primarily extending continuously for 30 km starting from the south margin of Eling Lake (Figure 3d) and almost west-trending. It was supposed to be the strongest deformed section both confirmed by the InSAR inverted result [6] and our field observations (Figure 4). The Eling segment's rupture is mostly single-stranded, showing obvious lateral slip-strike faulting with a strong compressive feature (Figure 4). The rupture developed on the frozen meadow with clay-rich surficial materials that dominate the landscape along the westmost area of the Jiangcuo Fault, and was defined by distinct continuous folded mole tracks, left strike-slip faults, right-stepping en echelon tensile fissures, fault scarps, and pressure ridges (Figure 4a–h); additionally, the ground was broken into turf rafts bounded by a series of en echelon faults. The multiple mole tracks coincided with rounded to angular folds of a deformed surficial layer, decoupled from its substratum. The cracks have individual lengths reaching tens of meters and widths that can be up to ~2.0 m (Figure 4d,f). The sand liquefactions (Figure 4h) were observed around the ponds close to the rupture zone. The rut prints and paths on the grassland were dislocated by left-lateral strike-slip activity. The co-seismic horizontal displacements along this section are in the range of ~0.4—2.6 m (Figure 3e). In this area, there was no obvious geomorphic trace of the seismogenic fault identified from the image before the earthquake, which indicated that this segment of the fault had developed recently.

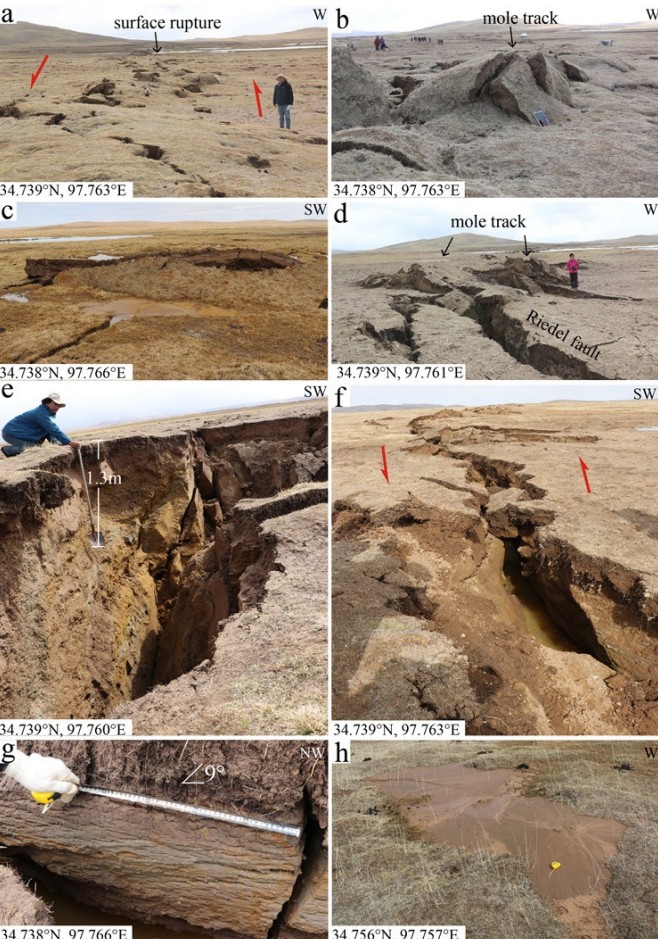

**Figure 4.** Representative field photographs of the rupture trace on the South Eling Lake segment: (**a**) continuous up-bulged mole track on the folded rupture; (**b**) mole tracks developed on a folded section; (**c**) typical pressure ridge; (**d**) up-bulged mole track bounded by rotated Ridel faults; (**e**) extensional crack; (**f**) continuous distributed extensional crack along the surface rupture; (**g**) fault slickensides; (**h**) sand liquefaction.

Yematan segment: The Yematan segment runs continuously along the east of the Eling Lake, showing clear continuous linear features on the DSM (Figure 2b), extending eastward across the north of Jiangcuogongma Lake and Jiangduo village, further to the Yematan Bridge (Figures 3d and 6a,b), and generally developed within alluvial deposits and intertwined with a concreted road along the mountain-range front, with a length of ~30 km. Three typical sites with different deformation styles were selected from west to east that displayed the characteristics of a co-seismic surface rupture.

The Jiangcuogonama Site: The rupture here is characterized by a left strike-slip and partial thrust movement, marked by continuous compressive ridges, multiple right-stepping en echelon tension fissures, and shear tension fractures (Figure 5A). The tension fissures display a complex morphology, such as an orthogonal shape, and X shape within the main deformation zone ranging in width from several meters up to 200 m. The continuous sharp compressive ridge caused the obvious vertical height of the ~0.5–1.5 m fault scarp (measured on DSM) and shows the obvious left-lateral strike-slip movement (Figure 5a–d).

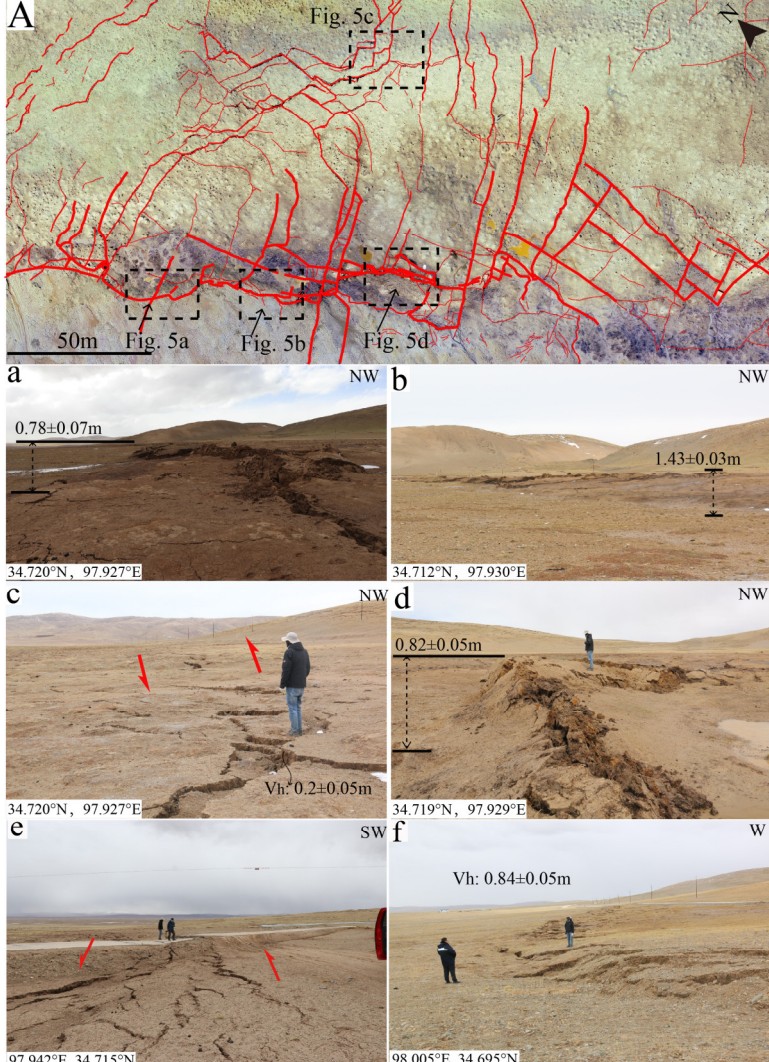

**Figure 5.** Surface rupture at Jiangcuogongma (**A**, **a–d**) and Jiangduo village (**e,f**) sites: (**a,b**) continuous pressure ridges; (**c**) right-stepping en echelon extensional cracks; (**d**) pressure ridge on the partial thrust fault; (**e**) extensional cracks developed on rupture zone with about 0.2–0.4 m vertical scarp; (**f**) right-stepping en echelon extensional cracks; here, the south side is down-dropped relative to the north side to form a scarp. (**A**) representative orthophotos of surface rupture on Jiangcuogongma site and the locations of photos (**a–d**).

The Jiangduo village site: The Jiangduo village site showed sinistral strike-slip faulting with an obvious component of normal faulting (Figure 5e,f). Overall, it strikes NWW 150°, and the surface rupture exhibited the characteristics of alternating ridges and fully separated cracks and fissures in a relatively narrow zone from several meters to about 30 m. Along the rupture, widely distributed en echelon right-stepping sub-segments vary in length from several meters to more than 20 m. The maximum co-seismic vertical displacement measured from the field was ~1.5 m. The normal component offset gradually decreased eastward, indicating the fault activity was gradually dominated by a sinistral strike-slip motion.

The Yematan Bridge site: The Yematan Bridge was the most devasted site in the Maduo earthquake. The rupture just crossed the north of the Yematan Bridge, leading to a directional drop off like chains of dominos (Figure 6b). The surface rupture zone near the bridge was about 100 m wide with 3–4 parallel cracks; the cracks had individual lengths reaching tens of meters and widths up to 10 cm, with a west–east orientation (Figure 6c,d). The west segment of the co-seismic surface rupture of the Yematan Bridge was characterized by a series of continuous right-stepping en echelon faults (Figure 6a) while the east of which was mainly characterized by shear faults and en echelon tensional cracks (Figure 6c,d), indicating that the east of the Yematan Bridge mainly demonstrates the characteristics of a pure left-lateral strike-slip.

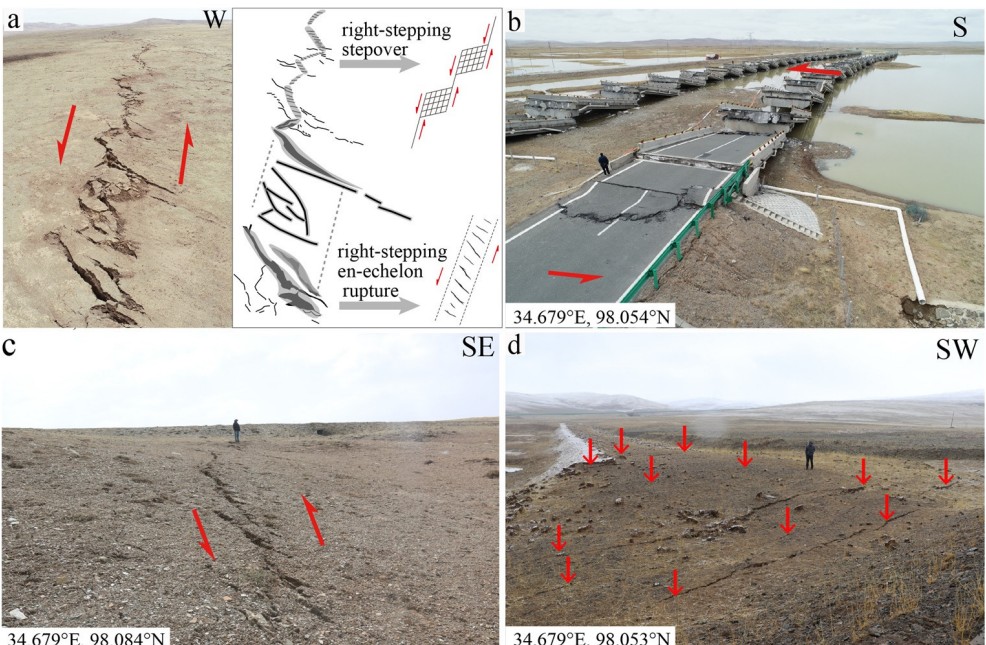

**Figure 6.** Representative photos of rupture at Yematan Bridge: (**a**) orthophoto and the geometry schema of right-stepping en echelon rupture close to Yematan Bridge; (**b**) collapsed Yematan Bridge; (**c**) right-stepping echelon fissures; (**d**) multiple parallel extensional cracks along the rupture close to the Yematan Bridge.

Huanghe town segment: The co-seismic rupture of the Hunghe village segment was mainly distributed continuously along the concreted road from the east of the Yematan Bridge to the Huanghe Township with a length of about ~30 km and a general strike that was north-west trending. Within this section, the fresh surface rupture was distributed discretely, as the rupture runs through ponds, water meadows, and sand dunes, which appear to partly absorb the stress strain. Unlike the western segment's widely developed extrusion bulges and compressive ridges, a number of discontinuous echelon extensional cracks, shear subfaults, and parallel or shear fissures were widespread presented within this segment. (Figure 7a–c,A,B). Farther east, the slip gradually dies out towards its termination in a small intermontane basin about 20 km eastward of Huanghe town.

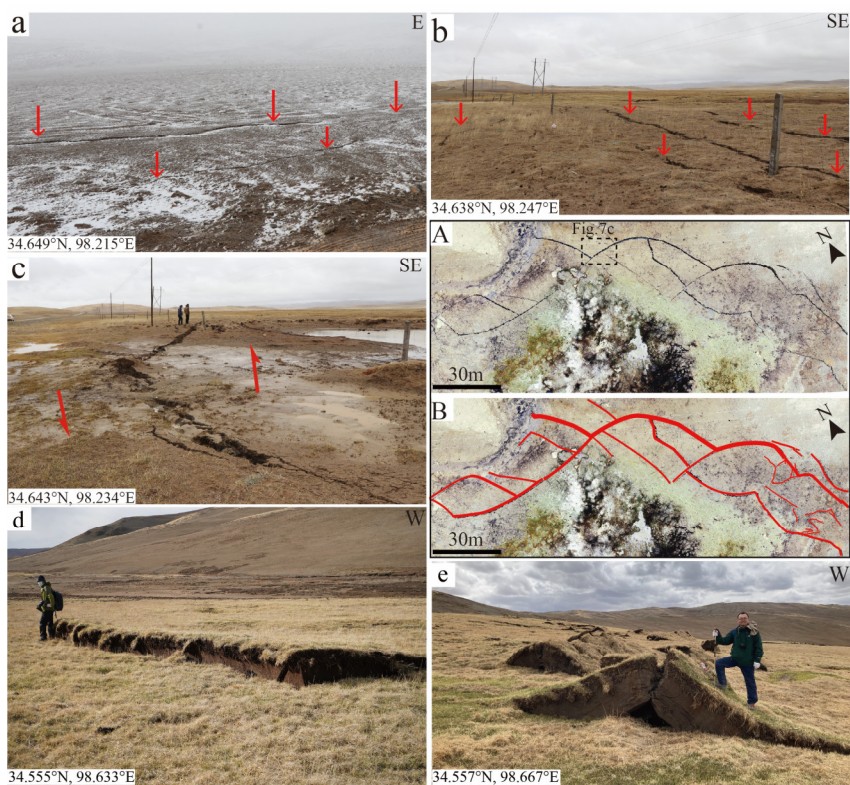

**Figure 7.** Surface rupture along the Huanghe town segment (**a**–**c**, **A**,**B**) and Dongcaoalong lake segment (**d**,**e**); (**a**–**c**) multiple extensional cracks on the surface rupture; (**d**) partial compressional thrust scarp; (**e**) bilateral symmetrical mole tracks developed along the rupture. (**A**,**B**) Representative orthophotos of rupture showing shear extensional cracks.

Dongcaoalong lake segment: This segment is left-stepped and separated from the Hunaghe town segment by a small pull-apart basin. The total length of this segment is longer than 45 km, with a SE strike, and is mainly interruptedly distributed along the north of Dongcaoalong lake. The entire rupture displayed an S shape; the longest continuous rupture was up to >5 km, located near the tensional bending part of the S-shape, characterized by multiple tensile fractures and extrusion bulges as high as 1 m (Figure 7d–e). The magnitude of the rupture declined on both sides, with scattered tensional cracks and fissures. Further east of the Dongcaoalong lake, a 10-km-long section showed no signs of any surface rupture. This rupture gap was probably due to the soft sediment effect of the site, which is spatially correlated with the sand-dune terrain that absorbed the earthquake deformation.

Changmahe segment: This segment was the easternmost tip of the rupture, mainly distributed in the north of Xuema town, with a strike of ~180°. About 2 km eastward, the main fault splays from a narrow discrete fault zone into a horsetail pattern with two sub-faults bending to the south (Figure 3d). The surface rupture broke along the north main fault with a total length of ~30 km, mainly consisting of en echelon tensional cracks and partially of minor extrusion bulges. Due to the difficulty in accessing the water and swampy ground further eastward, we could not trace the surface breaks beyond the east of Xuema town.

### 4.3. Co-Seismic Horizontal Displacements

Based on the field measurements and high-resolution DSM dislocation recognition, 42 horizontal displacements and 56 vertical displacement values of the Maduo earthquake have been summarized in Figure 3e, and the typical displacements for the different segments are shown in Figures 8–14. The sinistral horizontal displacements varied from several centimeters up to ~2.6 m, most of which were in the range of 0.5–1.6 m. Along the western segment of the surface rupture, the horizontal displacements exhibited two high peaks separated by a distinctly low region on the rupture bend. The maximum horizontal displacement measured

onthe south of Eling Lake was 2.6 ± 0.2 m. The vertical offsets were lower and distributed unevenly, with heights from a few centimeters to a maximum of 1.04 ± 0.05 m (Figure 3e) and most were below 0.5 m. Most of the vertical displacements were concentrated in the South Eling Lake segment and the Yematan segment, while no obvious vertical displacements were identified in the Huanghe village segment (with the vertical values of zero in Figure 3e). Figure 3e also shows that the vertical movement on the two ends of the surface rupture (the south Eling Lake and Changma river segments) were predominantly south-side up while the other parts were mainly north-side up. The typical horizontal displacements obtained both in the field and from the orthoimages in different segments are listed in Figures 8–12.

Typical horizontal displacement sites along the south of Eling Lake segment. Based on the interpretation of the orthoimages, Figure 8a–e show the typical dislocated tire tracks (by 1.4 ± 0.15 m), rut prints (by 1.5 ± 0.1 m; by 1.3 ± 0.1 m), and a pool bank (by 2.0 ± 0.15 m) on the western south Eling Lake rupture. Pan et al. [36] measured the dislocated long tire track lines (Figure 8a) at the south Eling Lake site and obtained left strike-slip displacements of 2.9 m, while we determined the close dislocated tire trend lines to have sinistral displacements of 1.4 ± 0.15 m (Figure 8a,B). The discrepancy was due to the uncertainty of the straight long tire trace; hence, the displacement of 1.4 ± 0.15 m was determined to be the minimum reliable displacement.

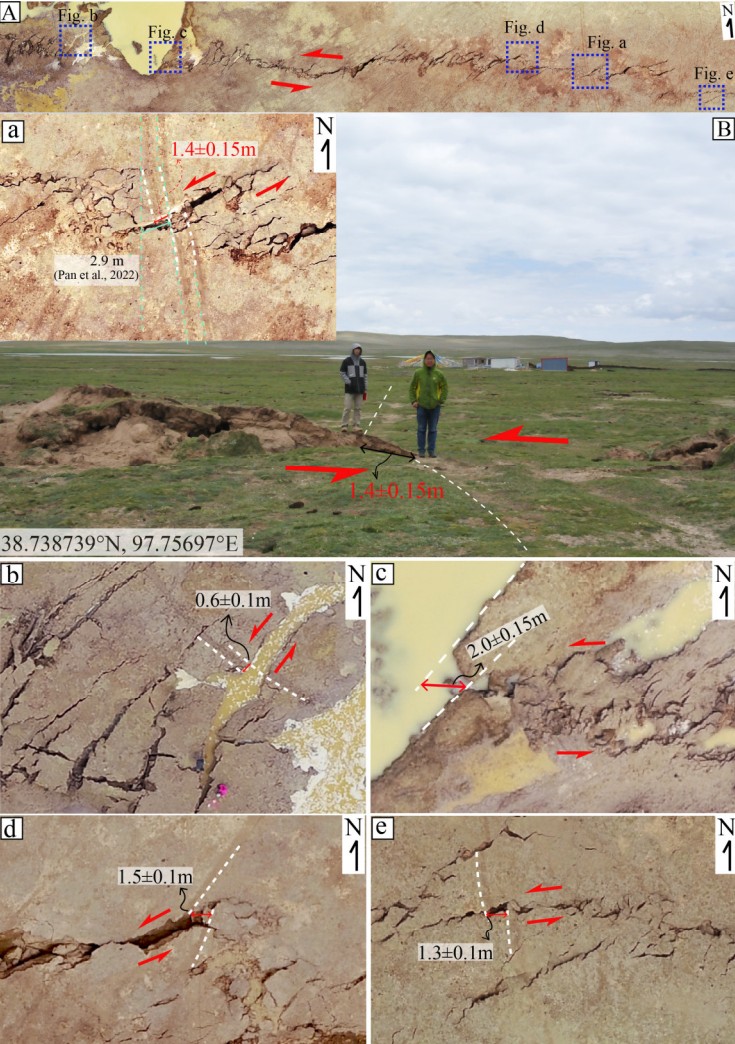

**Figure 8.** Orthophotos of co-seismic left-lateral strike-slip displacements on the South Eling Lake segment. (**A**) Index locations of offsets **a**–**e**; (**a**,**B**): orthophoto and field photo of tire tracks dislocation; (**b**): left-lateral offset of a rut print; (**c**): left-laterally dislocated bank of a small lake; (**d**,**e**): left-lateral offsets of pathways on the grassland.

The region of Figure 9A is 1 km to the east of the site of Figure 8e along the South Eling Lake segment. The continuous rut prints were offset by the main fault trace and multiple echelon sub-faults in the stepover zone. The sinistral offset of the rut prints was 0.7 ± 0.1 m (Figure 9a). The main fault displaced the tire tracks by 1.6 ± 0.1 m and the oblique-arranged sub-fault displaced the lower tire tracks by 0.4 ± 0.1 m (Figure 9B,c). The open cracks formed by the faults could reach up to 1.5 m wide in this area (Figure 9B), indicating the more intensive regional strain in the stepover zone along this segment of the rupture.

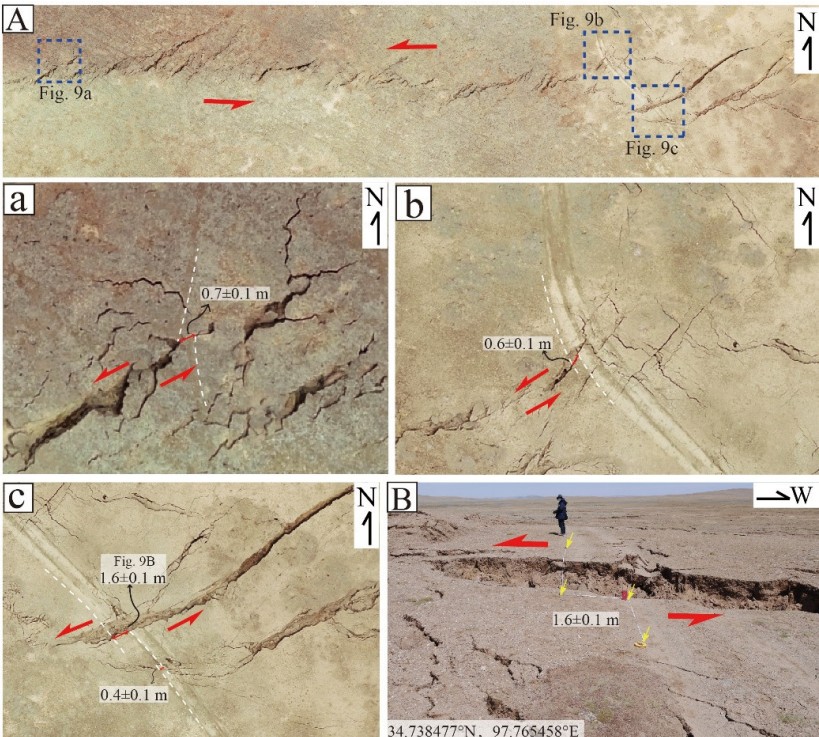

**Figure 9.** (**A**): Orthophoto of surface rupture and left-lateral offsets along the south Eling Lake; (**a–c**): left-laterally displaced rut prints on the grassland; (**B**): field photo showing left-lateral displaced rut prints on the grassland of the site on (**c**).

Typical displacement sites in the Yematan segment. Figure 10a–c show the typical deformable concrete road in the earthquake's path along the Yematan segment. In Figure 10a, the road was pushed up with a horizontal overlapping distance of 3.7 m, while Figure 10c indicates the two levels of left-lateral displacements offset by 0.39 m and 0.41m. Figure 10b shows the extrusion bulge of the concrete road on the surface rupture. Although the dislocation of the concrete road could not be regarded as a true co-seismic displacement due to the discontinuous deformation, it could indicate the rupture style and the referenced displacements. Four pathways on the grassland west of the Yematan Bridge were left-laterally displaced by 0.7 ± 0.1 m, 1.1 ± 0.1 m, 0.5 ± 0.1 m, and 1.5 ± 0.15 m (Figure 10A). Figure 10B,D show that the tire tracks were left-laterally displaced by 1.9 ± 0.2 m (Figure 10B) and 1.7 ± 0.1 m (Figure 10D) in two sites west of the Yematan Bridge.

Displacements along the Huanghe village segment. There were few displacement marks identified in this segment. The only obvious horizontal displacement observed consisted of the three levels of left-laterally displaced concrete road by 0.20 ± 0.05 m, 0.40 ± 0.1 m, and 0.15 ± 0.05 m. The total left-lateral displacement was 0.75 ± 0.15 m (Figure 10d).

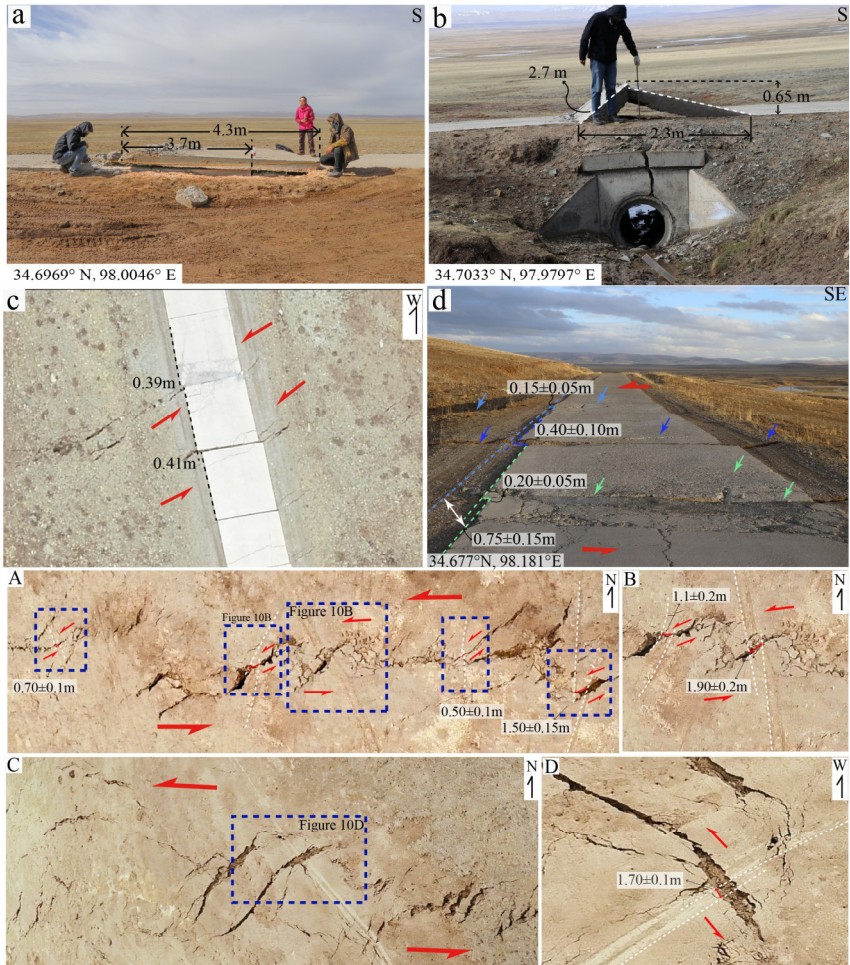

**Figure 10.** Representative field photos (**a–c**) and orthophotos (**A–D**) of co-seismic left-lateral strike-slip movement and offsets along the middle to east surface ruptures. (**a**): Compressional concrete road—the north side of the road was thrust onto the south side; (**b**): Rupture broke the road, forming a mole track; (**c**): UAV image showing that the paved road was displaced left-laterally by ~0.39 m and ~0.41 m; (**d**): Field photo of multiple-levels of left-laterally displaced concrete road near Huanghe village; (**A**,**C**): Orthophotos of left-laterally displaced rut prints and tire track; (**B**) Left-laterally displaced tire track in (**A**), which was displaced by 1.90 ± 0.2 m. (**D**) Left-laterally displaced tire track in (**C**), which was displaced by 1.70 ± 0.1 m.

### 4.4. The Shortening of the Compressive Bulges along the Co-Seismic Surface Rupture of the South Eling Lake Segment

Along the South Eling Lake segment, ~15 km long continuous remarkable mole tracks developed on the folded ridge, making it the most strongly deformable section (Figures 4 and 11); the horizontal shortening caused by the extrusion bulges and mole tracks was non-negligible. Zhao and Qu (2021) integrated interseismic and co-seismic geodetic observations to quantify the interseismic strain rate and co-seismic slip distribution, revealing that the maximum north–south displacement occurred in this segment, and the total strain intensity with the largest magnitude was particularly noticeable on the west segment of the rupture [6]. Based on the high-resolution DSM (0.03 m/pixel), 15 selected bulge profiles were extracted along the most deformable surface rupture, and the horizontal shortening rate was calculated from the projected length subtracted from the top length of a bulge. Figure 11 displays the typical bulges selected for preparing the bulge profiles and horizontal shortening.

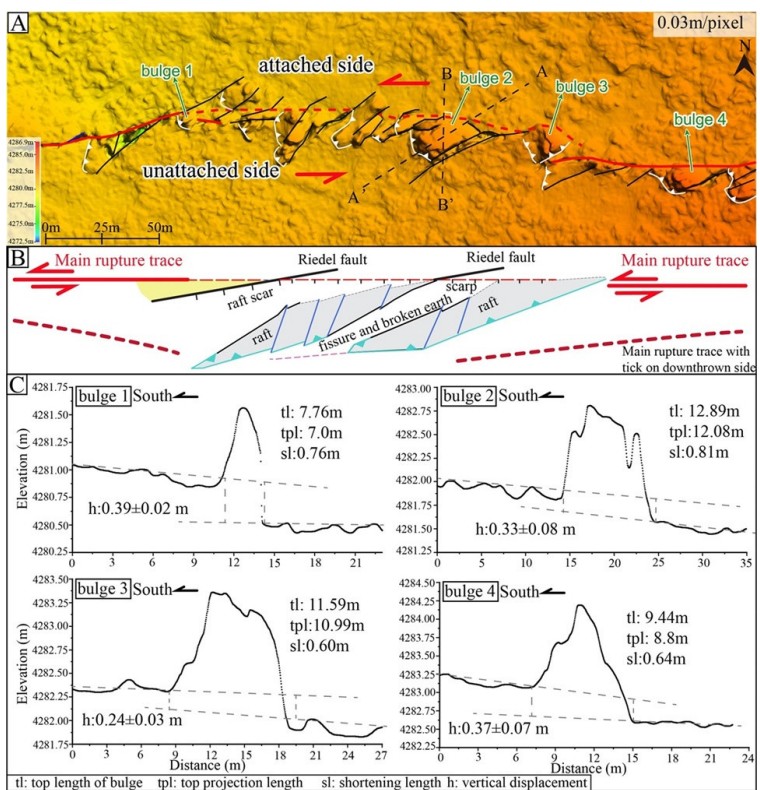

**Figure 11.** (**A**) High-resolution DSM of bulges in the South Eling Lake rupture segment. (**B**) Simplified diagram of mole track. AA'is the profile of the long axis of the bulge and the BB'is the profile perpendicular to the fault trace through the bulge vertex. The rafts are relatively pinned from one fault block "attached side" (northern block) and "unattached side" (southern block) to the other, from which they are thrust outward along a contractional fault. (**C**) Figures showing the bulge profiles extracted from DSM, the measurement of vertical scarp height, and the calculated horizontal shortening.

Figure 12 shows the spatial distribution of the co-seismic displacements and the shortening of the bulges measured from the DSM. The horizontal shortening lengths range from 0.2 m to 0.9 m. The maximum shortening displacements were distributed in the stepover zone of the two surface rupture strands with a maximum horizontal shortening of 0.9 m. The sinistral displacements within this section range from 0.3 m to 2.6 m, with a vertical component of 0.3–0.5 m, and the northside being downthrown.

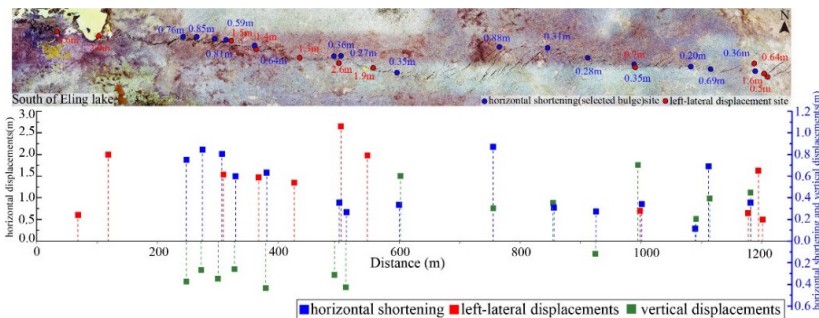

**Figure 12.** The distribution of horizontal shortening (calculated from the bulges), left-lateral displacements, and vertical displacements along the west of the south Eling Lake segment.

### 4.5. Cumulative Displacements Associated with the Co-Seismic Surface Rupture

Along the Yematan segment, the rupture broke eastward along the front of the mountain, with prominent triangular facets that are tens of meters high; the trace of the co-seismic rupture is also very distinctly present on the high-resolution DSM, which was

superimposed on the pre-existing fault scarps. The typical landforms, such as systematically deflected stream channels and fault scarps, can be noted along the main fault trace (Figures 13 and 14). Within this segment, we measured the cumulative offsets of previous seismic events where there were linear surface markers such as different levels of terrace risers and streams (Figures 2b–c, 13 and 14).

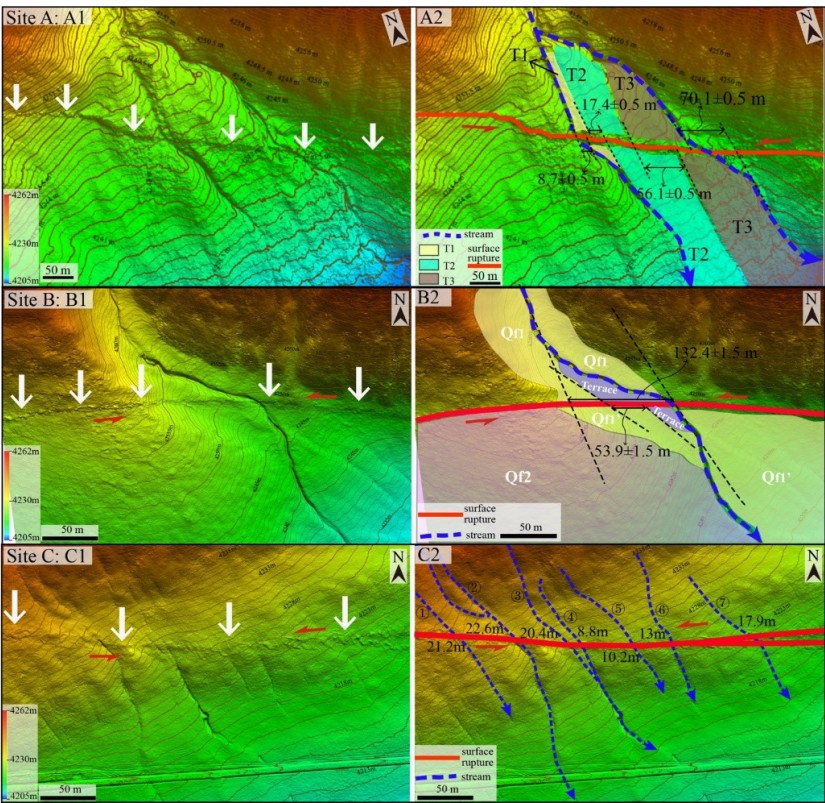

**Figure 13.** Co-seismic surface rupture broke along the preexisting fault where terraces and streams were sinistrally and systematically displaced. (**A1–C1**) are high-resolution DSM images of displaced geomorphic units; (**A2–C2**) are the geomorphic interpretation of figures (**A1–C1**).

Northwest of Jiangduo village, the rupture break runs along a large preexisting fault trough, where the stream and terraces of T1–T3 were sinistrally and systematically offset (Figure 13A1). The measured results show that the offsets of the terrace edge of T1-T3 are 8.7 ± 0.5 m, 17.4 ± 0.5 m, and 56.1 ± 0.5 m. The nearby stream was left-laterally displaced by 70.1 ± 0.5 m (Figure 13A2). At 1 km farther east from site A (Figure 13B1), the cumulative sinistral offset of a stream of about 132.4 ± 1.5 m was associated with the earlier displacement (Figure 13B2). At 1 km eastward of site B, a series of young displaced and deflected gullies also recorded the offsets of past earthquakes (Figure 13C1). The DSM measurement of the offset channels yielded a cumulative horizontal displacement range from 8.8 m–22.6 m for different levels of gullies (Figure 13C2).

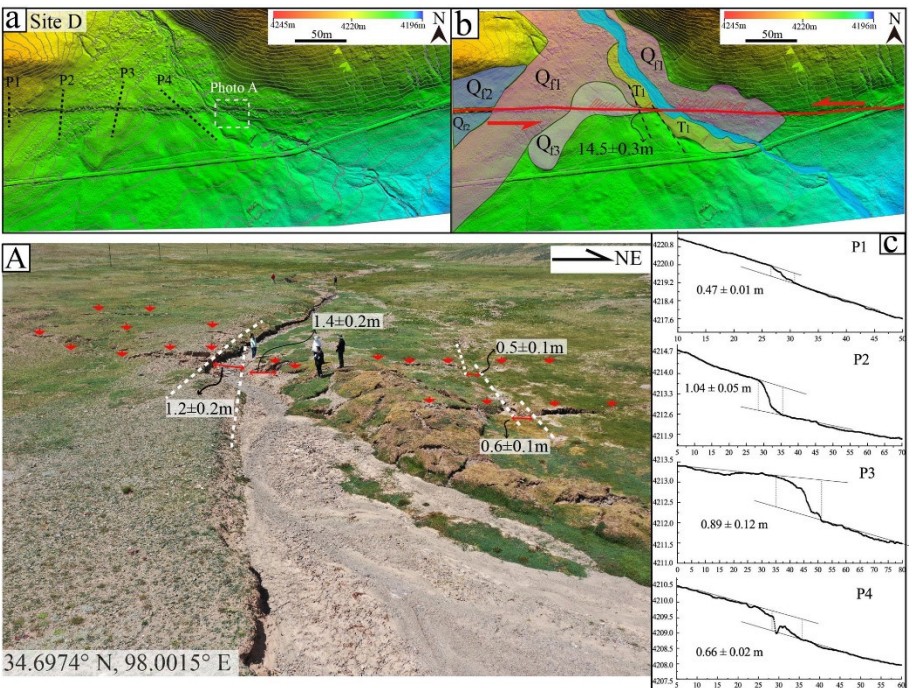

**Figure 14.** Co-seismic surface rupture along the preexisting fault where both co-seismic displacements and cumulative displacements were observed. (**a**) High-resolution DSM image of displacements at the observed site. (**b**) The geomorphic explanation of (**a**). (**c**) The profiles (p1–p4) show different levels of co-seismic fault scarp heights on alluvial fans (Qf1–Qf3) and on terrace T1. (**A**) Field photo of left-lateral displacements of the stream banks (location index in (**a**)) by 1.2 ± 0.2 m (southwest bank) and 1.4 ± 0.2 m (northeast bank), and small gullies nearby were dislocated by 0.6 ± 0.1 m and 0.5 ± 0.1 m.

About 1 km east of site C, both the co-seismic displacements and the cumulative offsets were observed from deflected streams and gullies (Figure 14). The stream was deflected by about 250 m and the relevant Holocene terrace riser of T1 was left-laterally offset by ~14.5 ± 0.3 m (Figure 14b). The northeast bank and the southwest bank of the stream were left-laterally displaced by 1.4 ± 0.2 m and 1.2 ± 0.2 m, while two gullies nearby were displaced by 0.5 ± 0.1 m and 0.6 ± 0.1 m, respectively, by the Maduo earthquake measured from the field (Figure 14A). Moreover, four topographic profiles (P1–P4) of four different alluvial fans from young to old were extracted from the high-resolution DSM, yielding the vertical displacements of 0.47 ± 0.01 m, 0.66 ± 0.02 m, 0.89 ± 0.12 m, and 1.04 ± 0.05 m (Figure 14c). All these systematically and progressively displaced terrace risers and gullies indicated the repeated activity of the fault during the Late Quaternary or the Holocene.

## 5. Discussion

### 5.1. The Seismic Gap on the Kunlunshankou-Jiangcuo Fault

The 2021 Maduo earthquake was distributed along the NWW-trending Jiangcuo Fault. The west end of the Jiangcuo Fault extending westward could be geomorphically connected to the Kunlun Mountain pass, where the Mw 7.8 Kunlun earthquake occurred in 2001. Hence, these two segment faults were collectively referred to as the Kunlunshankou-Jiangcuo Fault (KLSK-JCF), which is a large secondary fault belonging to the East Kunlun mega-fracture zone. For the entire KLSK-JCF, the Mw 7.8 Kunlun earthquake ruptured the westmost segment of the KLSK-JCF with a 450 km long surface rupture, while the Mw 7.4 Maduo earthquake ruptured the extreme eastern segment of the KLSK-JCF with a ~160 km long rupture (Figure 15). Between these two segments of ruptures, there is still a more than ~240 km seismic gap on the KLSK-JCF.

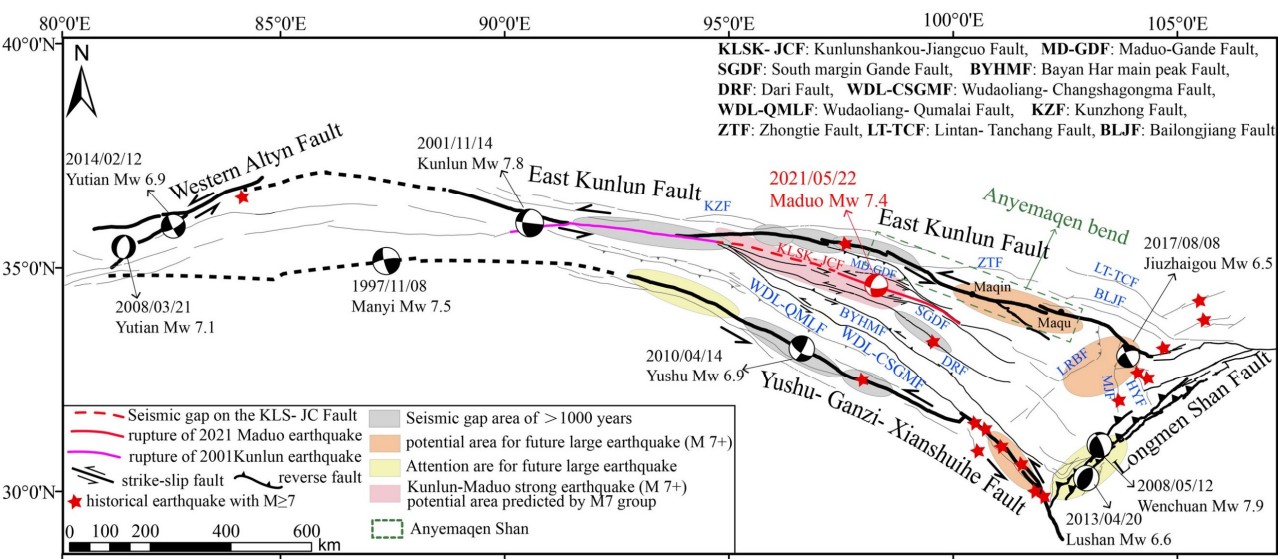

**Figure 15.** The boundary active faults and major large earthquakes as well as the evaluated potential areas of large earthquakes surrounding the Bayan Har Block, modified from (Working group of M7, 2012) [37] and (Pan et al., 2021) [38]. The pink line and red line indicate the rupture of the 2001 Mw 7.8 Kunlun earthquake and 2021 Mw 7.4 Maduo earthquake, respectively, while the dotted red line is a seismic gap without a rupture as of yet.

*5.2. The Tectonic Implications of the Bayan Har Block Indicated by the Maduo Earthquake*

Along the northern boundary of the Bayan Har Block, the East Kunlun Fault, a pronounced eastward decrease in the left-lateral displacement rates from > 10 mm/yr to < 2 mm/yr near the eastern fault terminus [20,21,24] was widely observed. Recent studies have suggested that this slip-rate gradient may be accommodated by block's' internal absorption and the regional clockwise rotation of the Kunlun Fault [39]. The differential eastward extrusion between the Songpan-Ganzi block and the Qaidam block as well as the sinistral slip across the Eastern Tibetan Plateau lead to a group of "horsetail" sub-faults at the east end of the East Kunlun Fault zone (Figures 1 and 15). More strikingly, the trace of the East Kunlun Fault exhibits a marked bend near ~98°E of the Anyemaqen Shan, with a fault strike nearly E–W to the west, whereas to the east, the fault strike changes to ~110° (Figure 15). This high-topography and thickened crust, region-scale bend prevented the continuous eastward propagation of the rupture, thus forming a series of sinistral strike-slip brunches. That is, the lateral shear of the Bayan Har Block driven by the eastward extrusion of the Tibetan plateau produces a cluster of wrenching-style sub-faults close to the east end of the East Kunlun Fault zone. The termination of the Kunlun fault zone is accommodated by a combination of distributed crustal thickening and by a clockwise rotation of the multiple eastern sub-faults [39,40]. Hence, it is suggested that multiple NWW trending faults inside of the Bayan Har Block near the Anyemaqen bend area, including the Xizang-Changmahe Fault, the Maduo-Gande Fault, the Kunlunshankou-Jiangcuo Fault, the South fringe Maduo Fault, and the Dari Fault (Figure 15) could be regarded as the sub-faults of the East Kunlun Fault. These faults initiated from the East Kunlun Fault zone become a broad "horsetail-like" brunch fault zone to adjust and transfer the eastward strain of the Bayan Har Block. It is quite different from a rigorous rigid block such as the Qaidam, Ordos, and Yangtze blocks. The main East Kunlun Fault and these sub-faults constitute a broad and dispersive northern boundary of the Bayan Har Block, controlling the inner strain distribution and deformation of the block, which is also corroborated by the study of the deep structure inversion of the Maduo earthquake area [41].

Even though the northern boundary of the Bayan Har Block exhibits broad, dispersive characteristics, the block itself was still regarded as relatively rigid. There have been eight strong earthquakes (1997 Manyi, 2001 Kunlun, 2008 Yutian, 2008 Wenchuan, 2010 Yushu, 2013 Lushan, 2014 Yutian, and 2017 Jiuzhaigou earthquakes) with magnitudes

above Mw 6.5 that occurred along the boundary faults of the block since the year 1995 (Figure 15), which have dominated the major strong earthquake cluster in the past 20 years on the Tibetan Plateau [42]. Unlike those earthquakes, the Maduo Mw 7.4 earthquake occurred within the Bayan Har Block; however, according to our analysis, it is still related to the eastern northern boundary system of the Bayan Har Block. The occurrence of the Maduo earthquake also implies that the deformation of the Bayan Har Block is under the mutual and coordinated control of mega-boundary faults with a high strike-slip rate and multiple secondary faults with relatively slow strike-slip rates. Hence, concern should also be directed to the southern boundary of the Bayan Har Block when considering the strain transferred and released along the "relatively rigid" dispersive boundary.

*5.3. Seismic Hazard around the Bayan Har Block*

The occurrence of the Maduo 7.4 earthquake on a young and relative minor fault [35,43] has led to an increasing amount of attention to the occurrence of large earthquakes within the Bayan Har Block. The historical earthquake record demonstrates that large earthquakes have usually occurred on the boundary faults of major blocks [42]. Based on the comprehensive analysis of historical earthquakes and the behavior of active faults, seismologists (Working group of M7) in China forecasted the mid-to-long-term potential of strong earthquakes on the Chinese Continent years ago [37] and suggested that the Kunlun-Maduo region was an M ≥ 7.0 potential strong earthquake area during the years from 2010–2020. Along the northern boundary of the Bayan Har Block (East Kunlun Fault), there are still two seismic gaps without any historical earthquake record: the Xidantan-Dongdatan and Maqin-Maqu segments [44]. The average recurrent intervals of the paleo-earthquake events are 600a and 1000a in these two segments, whereas there has been a 500a and 1000a lapse since the last strong earthquake in those two segments [45,46]. It has been very close to the average recurrent interval of M ≥ 7.0 earthquakes. Therefore, it can be inferred that the Maqin-Maqu segment has accumulated much more strain energy, and has a high potential for M7+ earthquakes, which is of great concern (Figure 15).

Along the southern boundary of the Bayan Har Block, the middle segment of the northern Xianshuihe Fault shows the long-term vertical uplift deformation, Moreover, a seismic gap still exists between the Kangding and DaoFu section on the middle segment of the Xianshuihe Fault [37]. Hence, the middle segment of the Xianshuihe Fault remains a potential source for a large M7+ earthquake in the next 10 or more years. According to the seismic gap principle, special attention should also be paid to the south segment of the Longmenshan Fault and Longriba Fault (Figure 15) for their increased earthquake potentials with respect to the southeast boundary of the Bayan Har Block [37].

## 6. Conclusions

(1) The 2021 Mw 7.4 Maduo earthquake produced an NWW-trending ~160 km-long surface rupture stretching from south Eling Lake in the west to Changmahe town in the east. The earthquake occurred as a result of the predominantly left-lateral strike-slip faulting with a component of normal dip-slip. The seismogenic fault is the Kunlunshankou-Jiangcuo Fault, which ruptured the Jiangcuo Fault segment (east segment of KLSK-JCF) in the 2021 Maduo earthquake.

(2) The co-seismic surface rupture mainly consists of distinct shear faults, right stepping en echelon tensional cracks, mole track structures, and widely distributed water blasting, sand liquefaction, and earthquake pits. The maximum sinistral strike-slip displacement is ~2.6 m and the maximum vertical displacement is ~1.5 m.

(3) The deformation of the Bayan Har Block is under the mutual and coordinated control of the main East Kunlun Fault and the other six NW-trending left-lateral strike-slip subfaults (the Maduo-Gande Fault, the Kunlunshankou-Jiangcuo Fault, the Gande south margin Fault, the Dari Fault, the Bayan Har main mountain Fault, and the Wudaoliang-Changshagongma Fault). The main East Kunlun Fault together with these subfaults constitutes a broad and dispersive northern boundary of the Bayan

Har Block. The Mw 7.4 Maduo earthquake indicated that the Bayan Har Block still has the potential to produce strong earthquakes of M7+. The Maqin-Maqu segment on the northern boundary, the middle segment of the Xianshuihe Fault, and the Longriba Fault area should be particularly scrutinized for their intermediate-term large earthquake potentials.

**Author Contributions:** field investigation, writing—original draft preparation, H.X.; field investigation, funding acquisition, Z.L.; writing—review and editing, D.Y., X.W., Q.S.; field investigation, X.L., A.W., P.S.; All authors have read and agreed to the published version of the manuscript.

**Funding:** This research was funded by "Key Research and Transformation Plan of Qinghai Province", grant number 2022-SF-138, and "National Natural Science Foundation of China", grant numbers 41302174 and 42172227.

**Data Availability Statement:** Not applicable.

**Acknowledgments:** We would like to express our gratitude to Sotirios Valkanioti who supplied the LiCSAR observations result of the 2021 Mw 7.4 Maduo earthquake, and the Public Service Department of Gansu Earthquake Agency for coordinating the scientific investigation of the Mw 7.4 Maduo earthquake. We also appreciate the constructive suggestions from editors and reviewers for improving the manuscript.

**Conflicts of Interest:** The authors declare no conflict of interest.

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
