# Peer review of "Characteristics of Co-Seismic Surface Rupture of the 2021 Maduo Mw 7.4 Earthquake and Its Tectonic Implications for Northern Qinghai–Tibet Plateau"

_remotesensing, doi:10.3390/rs14174154_

Round 1

Reviewer 1 Report

I’ve read the manuscript of subject and I found that is an acceptable ducumentation of the field measurements and investigations performed by the authors. The theoretical interpretation – like any theoretical interpretation – could be matter of discussion. The scientific cummunity preferes to discuss cases and interpretation options that are supported by field observations. Form this point of view the manuscript has a value, since it provides emporical evidences, even if somebody is questioning certain details of the research. I think my position does not need more detailed argumentation.

Reviewer 2 Report

The goal of the reviewed manuscript, as the authors declare, is the study of the coseismic surface rupture of the 2021 Maduo earthquake, on the Qinghai-Tibet Plateau. In my opinion, this paper needs to be largely improved before to be accepted for publication in the Remote Sensing journal, especially concerning the used methodology, as well as the clarification of some aspects along the work.

My main concerns are the following ones. First at all, this has been an earthquake largely studied, also its rupture, in several published papers, some of them in this journal. It is not clear at all to me what is innovative in this work, both in results and implications, to be published. That is, which results and conclusions shown here have not previously published? The second question is that, in my opinion, a work like this deserves to be published in a journal like Remote Sensing only if the used methodology falls within the field of the remote sensing science, if not, this is a manuscript that deserves to be sent to a journal on Geology or Tectonics. Reviewers of these type of journals are expert in these fields, and will better judge the results and inferences presented here. Well, the authors dedicate only a few lines (131-148) to comment, or even explain, the methodology used in this work. Then, I don’t found reasons to publish this research in this journal. Other question is the section 5.3. It is not clear to me the meaning of this section, more taking into account that authors use data not included in this work, i.e., data published previously by other authors. It has not sense to include this section, taking into account that to justify these conclusions imply to know data on paleo-earthquakes return periods, Coulomb stresses, historical earthquake events, etc., not related to this work, and not available for readers (PhD Thesis, papers in Chinese, …). Finally, many references of this paper are written in Chinese, then, as reviewer and potential reader, I cannot check the referred data, and I just have to blindly believe the authors. In a SCI journal this should not be allowed.

Other ‘minor questions’ are the following ones.

- First at all, the manuscript needs some English revision.

- The term ‘risk’ along the manuscript (earthquake risk) is misused. The correct term is earthquake or seismic ‘hazard’. Authors relate this term (risk) to the occurrence of events, not to the damage they can cause.

- References need an improvement. Some references included in the text are not included in the references section (e.g., Sotirios Valkanioti, 2021; M7 group, 2012, …), and some references in the references section are not included in the text (e.g., references 10, 11, 35, …). Several references do not show all authors (e.g., references 1, 2, 6, 9, 12, 14, …). There are two Yuan et al. (2022) works quoted in the references section. What is the one quoted in lines 291-292? By the way, one of these references is not referred. In addition, references (in the references section and in the text) do not follow at all the guidelines of this journal. All this does not give a good impression to the reader.

- Along the text and figures, repeatedly, authors join the magnitudes and the units (e.g., 1.5m, Mw7.4, 170km, <2mm/yr, …). Please, separate one from another (e.g., 1.5 m, Mw 7.4, 170 km, < 2 mm/yr, …).

- When referring the name of a fault, the word fault must be included in uppercase (e.g., East Kunlun Fault).

- Figures need some improvement, basically, they should be enlarged in order to observe clearly the different features, even showing some of them in portrait format instead of landscape format. Specifically, focal mechanism solutions in figures 1 and 15 must be enlarged, and the focal mechanism of the 2021 Maduo earthquake must be highlighted. To be a tectonic sketch plot, regional stress must be depicted. In addition, some figures are not referred in the text by order, e.g., the first quoted figure is the figure number 3. All them must be located near the place they are quoted for the first time.

- It must be clear in the manuscript that seismicity relocation, used to do some inferences by the authors, is not a result of this paper but a result previously published.

Reviewer 3 Report

This manuscript reports the detailed characteristics of coseismic surface rupture of the 2021 Maduo Mw7.4 earthquake. These results are important of understanding the physical processes of Maduo EQ. The writing is clear.

 Two comments:

Line 18: en-enchelon: en-echelon?

Line 21-33: I think, these parts are not the major contents, which are a potency of understanding  induced by auhors. These discussions are of great significance, however still  not be confirmed. I suggest this part, namely discussed in detail in section 5, should be reduced largely in abstract. 

Round 2

Reviewer 2 Report

My opinion on this new version of the manuscript is the same that with the first one: it is not suitable for publication in the Remote Sensing journal. I recognize that some minor changes/comments proposed by this reviewer have been considered, improving a little the writing of the paper, but not the most important ones.

My opinion I that this is a manuscript must be sent to a journal on Geology or Tectonics. Reviewers of these type of journals are expert in these fields, and will better judge the results and inferences presented here, but not experts on Remote Sensing technologies. They are that must judge what is innovative in this work, both in results and implications, more taking into account the large number of papers on this seismic event published previously, some of them in this journal. To be a paper on Remote Sensing approaches the authors dedicate only a few lines to succinctly explain the methodology used in this work.

              Conclusions of this work are clearly Geological and Tectonics aspects that must be justified from data on paleo-earthquakes return periods, Coulomb stresses, historical earthquake events, etc., not related to this work, and not available for readers, not even with the new added references.

Finally, the authors are not used to the terms seismic risk, seismic hazard and seismic potential. They continue misusing them throughout the manuscript.